# Ratios of Acetaminophen Metabolites Identify New Loci of Pharmacogenetic Relevance in a Genome-Wide Association Study

**DOI:** 10.3390/metabo12060496

**Published:** 2022-05-30

**Authors:** Gaurav Thareja, Anne M. Evans, Spencer D. Wood, Nisha Stephan, Shaza Zaghlool, Anna Halama, Gabi Kastenmüller, Aziz Belkadi, Omar M. E. Albagha, Karsten Suhre

**Affiliations:** 1Bioinformatics Core, Weill Cornell Medicine-Qatar, Education City, Doha P.O. Box 24144, Qatar; gat2010@qatar-med.cornell.edu (G.T.); nis2034@qatar-med.cornell.edu (N.S.); sbz2002@qatar-med.cornell.edu (S.Z.); amh2025@qatar-med.cornell.edu (A.H.); abb2013@qatar-med.cornell.edu (A.B.); 2Department of Physiology and Biophysics, Weill Cornell Medicine, New York, NY 10065, USA; 3Metabolon Inc., Morrisville, NC 27560, USA; aevans@metabolon.com (A.M.E.); swood@metabolon.com (S.D.W.); 4Institute of Computational Biology, Helmholtz Zentrum München, German Research Center for Environmental Health, 85764 Neuherberg, Germany; g.kastenmueller@helmholtz-muenchen.de; 5College of Health and Life Sciences, Hamad Bin Khalifa University, Doha P.O. Box 24144, Qatar; oalbagha@hbku.edu.qa; 6Centre for Genomic and Experimental Medicine, Institute of Genetics and Cancer, University of Edinburgh, Edinburgh EH8 9LH, UK

**Keywords:** medication, drug metabolites, population studies, non-targeted metabolomics, genome-wide association studies, acetaminophen metabolism, pharmacogenomics, 3-methoxyacetaminophen

## Abstract

Genome-wide association studies (GWAS) with non-targeted metabolomics have identified many genetic loci of biomedical interest. However, metabolites with a high degree of missingness, such as drug metabolites and xenobiotics, are often excluded from such studies due to a lack of statistical power and higher uncertainty in their quantification. Here we propose ratios between related drug metabolites as GWAS phenotypes that can drastically increase power to detect genetic associations between pairs of biochemically related molecules. As a proof-of-concept we conducted a GWAS with 520 individuals from the Qatar Biobank for who at least five of the nine available acetaminophen metabolites have been detected. We identified compelling evidence for genetic variance in acetaminophen glucuronidation and methylation by UGT2A15 and COMT, respectively. Based on the metabolite ratio association profiles of these two loci we hypothesized the chemical structure of one of their products or substrates as being 3-methoxyacetaminophen, which we then confirmed experimentally. Taken together, our study suggests a novel approach to analyze metabolites with a high degree of missingness in a GWAS setting with ratios, and it also demonstrates how pharmacological pathways can be mapped out using non-targeted metabolomics measurements in large population-based studies.

## 1. Introduction

Acetaminophen is a commonly used over-the-counter analgesic. It also has antipyretic effects. Known under the brand names Paracetamol and Tylenol, acetaminophen is the most frequently used painkiller world-wide. Acetaminophen is considered a nonsteroidal anti-inflammatory drug and thought to act via the inhibition of the cyclooxygenase pathway and prostaglandin biosynthesis [1]. Acetaminophen metabolism has been extensively studied in the past [2,3,4,5]. Acetaminophen is metabolized primarily in the liver and to a lesser extent in the kidneys and intestines [6]. At a therapeutic dosage, approximately 52–57% is converted to inactive glucuronide conjugates, 30–44% to sulfate conjugates, and 5–10% to the highly active metabolite N-acetyl-p-benzoquinone imine, which is then further detoxified through binding to the sulfhydryl group of glutathione, involving multiple enzymes from the UGT, SULT, and CYP gene families (https://www.pharmgkb.org/pathway/PA166117881, accessed on 15 April 2022).

Many genetic variants in genes involved in endogenous metabolism have been identified by genome-wide association studies (GWAS) [7]. In many cases, genetic variants in or near a metabolically active gene were found associated with metabolites that match the gene’s function, up to a point where it was possible to derive the gene’s function from the association profile [8], or to identify an unknown metabolite from its association with a specific enzyme [9]. Taken together, these variants are considered to define an individual’s metabolic individuality and can inform personalized medicine approaches and drug target predictions [10].

However, genetic variation in genes involved in exogenous metabolism and detoxification processes has been less studied so far, mainly because the relevant xenobiotics and drug metabolites are only present in small subsets of the study participants. This generally leads to their exclusion from GWAS due to their high degree of missingness, which typically requires a metabolite to be detected in at least 50% of the samples. A second challenge resides in the relative quantification of metabolites with high missingness: due to the large sample numbers required for GWAS, metabolomics measurements are required to be run in batches over a significant period of time, which can lead to differences between batches of several orders of magnitude, especially when changes in liquid chromatography columns are required or multiple instruments are used. Non-targeted approaches have several potential methods to estimate appropriate correction factors [11,12], which appears to work reasonably for metabolites with a low degree of missingness [13] but does not perform well when only a few data points per batch are available for normalization [14].

We previously used ratios between metabolite pairs in GWAS [8,10,15] and found that this approach could strengthen association signals significantly, partly because they implicitly correct for technical and biological variation, and partly because ratios approximate reaction rates between substrate and product pairs and therefore represent close proxies for enzyme activity, which is assumed to be affected by the genetic variants [16]. We also recently analyzed drug metabolites from non-targeted metabolomics and their overlap with self-reported medication in the Qatar Biobank (QBB) study [14], where we identified nine distinct acetaminophen metabolites that were detected together in over 520 study participants (Table 1).

We therefore hypothesized that genetic variance in acetaminophen metabolism could be identified by a GWAS on all possible ratios between these nine metabolites, and that the characteristics of the associated metabolite pairs would match the enzymatic function of the respective genes. In the following we describe what could be a prototype for the GWAS analysis of drug metabolites from non-targeted metabolomics and show that associations with ratios between related drug metabolites can reveal new information, both about the implicated metabolites as well as about the associated genes.

## 2. Results

### 2.1. A GWAS with Ratios between Acetaminophen Metabolites

For this study we used all samples of our previous study on medication use in QBB [14] for which at least five of the nine acetaminophen metabolites were reported (N = 520, Table 1). The participants in this group were aged between 18 and 76 years with an average age of 40.1 years, and 308 samples were from female study participants (59.2%). We tested the nine acetaminophen metabolites and all possible ratios between them (N = 45) for association with 6,375,079 genetic variants, using log-scaled metabolic traits as dependent variables and age, sex, and the first ten genomic principal components as covariates, largely following protocols previously described in our GWAS with biochemical chemistry traits in QBB [19] (see Methods).

Associations were considered as significant at a genome-wide and trait-wide level if their *p*-values were smaller than the multiple-testing corrected genome-wide significance threshold of *p*-value_Bonf_ = 1.1 × 10^−9^ (=5 × 10^−8^/45). In addition, we required a *p*-gain above p-gain_Bonf_ = 450 (=10 × 45). The *p*-gain of an association of a metabolite ratio is defined as the smaller value of the two *p*-values for a pair of metabolite associations divided by the *p*-value of the association for their ratio. A p-gain of 10 has been shown to be the equivalent of an alpha level of significance of 0.05 for a single test [16]. Large p-gains indicate pairs of metabolites that are linked through a common normalizing factor or a shared metabolic pathway.

The strongest association in our study had a *p*-value of 3.7 × 10^−10^ and a *p*-gain of 4.6 × 10^7^ for the association of SNP rs1531022 at the UGT2B15 gene locus with the ratio of 2-methoxyacetaminophen glucuronide* to 2-methoxyacetaminophen sulfate* and was therefore considered both significant at a genome- and trait-wide level and significant for the ratio. No other association reached a Bonferroni level of significance.

We therefore continued our investigation using a target gene approach. At a more liberal *p*-value of *p* < 10^−5^, we identified 229 associations at 217 independent loci (LD r^2^ < 0.1) and annotated these with the nearest genes and overlapping eQTL, pQTL, mQTL, and methQTL information using a PhenoScanner [20] and SNiPA [21] (Appendix A). We then searched the nearest gene annotations for 54 genes that are part of the acetaminophen metabolic pathway according to GeneCards or Drugbank (see methods, Appendix A). We identified three additional variants at the UGT2B15 locus and one variant near COMT (Table 2 and Figure 1).

### 2.2. The UGT2B15 Locus

Variant rs1531022 is an intronic SNP of the UDP Glucuronosyltransferase Family 2 Member B15 (UGT2B15) gene. UGT2B15 encodes a glycosyltransferase that is involved in the metabolism and elimination of toxic compounds from the body, including acetaminophen. The present association of a polymorphism in UGT2B15 with the ratio of a glucuronidated and a sulfated metabolite of acetaminophen, that is, methoxyacetaminophen glucuronide/methoxyacetaminophen sulfate, reflects a genetically determined shift in the contribution of the two detoxification pathways in the clearance of acetaminophen metabolites and matches the biological function of UGT2B15.

Polymorphisms in the drug metabolizing capacity of UGT2B15 have been reported as long as 25 years ago [22] and have been much studied since [23,24,25,26]. A highly prevalent non-synonymous SNP (rs1902023) is found in approximately 50% of Caucasians (g.253G>T, D85Y). It is a significant determinant of UGT2B15 interindividual variation and is found on the UGT2B15*2 and UGT2B15*5 haplotypes [27]. The correlation between SNP rs1902023 and rs1531022 is r^2^ > 0.99; therefore, rs1531022 can be considered as tagging this likely causal variant. The current association is therefore extending the experimentally established list of UGT2B15 substrates to methoxyacetaminophen.

We identified three additional, mutually uncorrelated variants (r^2^ < 0.1) at the UGT2B15 locus that did not reach genome- and trait-wide significance (Table 2). However, given their association with the same ratio as the rs1531022 variant and significant p-gains, these three variants are likely true positives, pending confirmation in future, more highly powered studies. Noteworthy is that two of these variants (rs72607822 and rs1968718) were also associated with an expression QTL for UGT2B11, which suggests that this gene may potentially also be involved in acetaminophen glucuronidation.

### 2.3. The COMT Locus

The catechol-O-methyltransferase (COMT) catalyzes the transfer of the methyl group of S-adenosyl-L-methionine to one of the hydroxyl groups of a catechol substrate (Figure 2). It is part of the degradative pathway of catecholamine neurotransmitters, such as dopamine [28,29]. SNP rs4680 is a common coding SNP (Val158Met) in COMT and has previously been associated with the unknown metabolites X-01911 and X-11593. The latter was subsequently identified as O-methylascorbate, which is a known product of vitamin C methylation by COMT [9]. rs4680 was reported to affect protein abundance and enzyme activity, but not its gene expression [29]. Two adjacent hydroxy-groups would make 3-hydroxyacetaminophen also a likely substrate of COMT, but not 2-hydroxyacetaminophen.

Our association of rs165722 (LD r^2^ = 0.98 with rs4680) with the ratio of 2-methoxyacetaminophen sulfate and 2-hydroxyacetaminophen sulfate and the biochemical function of COMT as a catechol-O-methyltransferase suggests that the hydroxy- and methoxy-groups are therefore on the three-position, and not in the two-position. Interestingly, there is literature precedence for both structures [18,30], but this GWAS analysis supports the identification as the three-form rather than the two-form. A genetic association of rs1531022 in UGT2B15 with vanillyl mandelate (VMA) has been reported previously [31]. The structure of VMA is very similar to that of 3-methoxyacetaminophen (3MA) (Figure 2), which lends further support to the hypothesis that the associated metabolites are 3-hydroxy- and 3-methoxy-forms of acetaminophen.

To confirm this GWAS-based hypothesis, the 3-isoform of the methoxy-acetaminophen metabolite was purchased and synthetically sulfated and glucuronidated. The data show that indeed 3-methoxyacetaminophen glucuronide and the 3-methoxyacetaminophen sulfate match the peaks detected in plasma (Figure 3), confirming the GWAS hypothesis. Future work includes synthesizing the 3-hydroxyacetaminophen sulfate and glucuronide molecules to determine whether they also match the peaks detected in plasma.

In summary, our associations suggest the following chain of reactions: (1) the conversion of acetaminophen to 3-hydroxyacetaminophen by a yet to be identified reaction step, (2) the conversion of 3-hydroxyacetaminophen to 3-methoxyacetaminophen by COMT, and (3) the sulfonation to 3-methoxyacetaminophen sulfate or glucuronidation to 3-methoxyacetaminophen glucuronide, with the branching of the final step being determining and under genetic control (at least in part) by the efficacy of the glucuronidation reaction catalyzed by UGT2B15.

### 2.4. Other Loci

We screened the list of associations for additional biochemically plausible cases and searched for loci that involve UGT, SULT, and CYP genes, and other genes that could be enzymatically relevant to acetaminophen metabolism. We found the following two potentially relevant loci:

SNP rs57945749 is located in an intergenic region at chr19:32616014 and was previously reported as a trans-eQTL (*p* = 6.3 × 10^−6^) with CYP2C8 [32]. The chemical properties of the two metabolites in the associated ratio, 3-(methylthio)acetaminophen sulfate* and 3-(N-acetyl-L-cystein-S-yl) acetaminophen, correspond to the function of CYP genes (*p*-value = 6.8 × 10^−6^, p-gain = 2911);

SNP rs2297813 near CYP4A11|CYP4A26P|CYP4B1|CYP4Z2P are associated with the ratio of 3-(N-acetyl-L-cystein-S-yl) acetaminophen and 4-acetamidophenol. While not explicitly annotated to the acetaminophen pathway, CYP4B1 participates in the metabolism of a number of aromatic amines [33], so it may be speculated that acetaminophen constitutes a possible substrate (*p*-value = 3.9 × 10^−6^, p-gain = 13,337).

## 3. Discussion

We conducted what is arguably the first GWAS with acetaminophen metabolites measured using the non-targeted HD4 metabolomics platform Metabolon Inc. (Morrisville, NC) and samples from the Qatar Biobank study. We present a novel approach to analyze metabolites with a high degree of missingness in a GWAS setting using ratios and show that pharmacological pathways may be gleaned by using data from large population-based studies. Our approach provides further insight into metabolomics-driven pharmacogenomics analyses of acetaminophen (Figure 4).

Specifically, the multiple associations that we reported at the UGT2B15 locus may now be used to support fine mapping of pharmacogenetic haplotypes at this important locus. Our results further suggest that 3-hydroxyacetaminophen may play a pivotal role in acetaminophen detoxification and add this molecule to the growing list of known COMT substrates.

As with all GWAS, our study has a number of limitations, most importantly its relatively low statistical power, and, as for now, a lack of an independent replication cohort. However, since the lead variants at the UGT2B15 and COMT loci have been previously associated with relevant biochemically related molecules, and since the top association reaches genome-wide significance even at the current number of samples, we do not feel that a lack of replication is an issue for these two loci. Regarding the discovery of further loci of interest, we consider this study as a proof-of-concept that may initiate the conduct of larger-scale GWAS with drug metabolites using a ratio-metric approach.

## 4. Materials and Methods

### 4.1. Study Population

Qatar Biobank (QBB) is a population study that includes adult Qatari nationals and long-term residents (≥15 years living in Qatar) [34,35]. QBB samples and data have been used in multiple genetic studies [19,36,37,38].

### 4.2. Metabolomics Measurements

A total of 3000 EDTA blood plasma samples were analyzed on the non-targeted metabolomics platform of Metabolon Inc. (Morrisville, NC, USA) at the Anti-Doping Laboratory–Qatar (ADLQ) as previously described [11,14]. Briefly, samples were analyzed on an LC–MS system, consisting of a Waters ACQUITY ultra-performance liquid chromatography (UPLC) unit and a Thermo Scientific Q-Exactive high resolution/accurate mass spectrometer, interfaced with a heated electrospray ionization (HESI-II) source and Orbitrap mass analyzer operated at 35,000 mass resolution. Three analyses were run using a C18 column, one of them with acidic positive ion conditions and two with basic negative ions, chromatographically optimized for more hydrophilic compounds and more hydrophobic compounds, respectively. A fourth analysis was run using negative ionization following elution from a HILIC column. Raw MS data were annotated by Metabolon using their proprietary in-house software and more than 3300 commercially available purified standard compounds as a reference. Peaks were quantified using area-under-the-curve. Data normalization was performed in run-day blocks by registering the medians to equal one and normalizing each data point proportionately. The overall relative standard deviation for instrument variability based on internal standards was 12% and the total process variability determined using endogenous metabolites detected in reference sample was 16%. The provided dataset comprised a total of 1159 biochemicals, 937 compounds of known identity (named biochemicals) and 222 unidentified compounds (unknowns). Full details of the QBB metabolomics data are provided in [14]. In this study we only use data from 520 QBB study participants for whom at least five of the nine acetaminophen metabolites were detected (see Table 4 in [14]).

### 4.3. Synthesis of 3-Methoxyacetaminophen Sulfate and Glucuronide Conjugates

Sulfation procedure: One equivalent of 3-methoxyacetaminophen (0.030 mmol) was added to 0.3 mL of pyridine in an 8 mL vial. Sulfur trioxide pyridine complex (5 mg, 0.33 mmol, 1.1 eq) was added to the vial and the mixture was stirred at 40 °C for 3 h. The reaction mixture was cooled to room temperature and 1N KOH (3.5 mL) was added to the vial dropwise. The resulting solution was transferred to a 20 mL vial and 14 mL of iPrOH was added and the solution was stored in a 5 °C chamber overnight. The liquid was decanted off the residue that formed overnight, and the resulting residue was dried with a stream of N_2_ gas and lyophilized to yield a sticky solid residue [39].

Glucuronidation Procedure: One equivalent of 3-methoxyacetaminophen (0.030 mmol) was added to 0.5 mL of benzene in an 8 mL vial. Acetobromo-α-D-glucuronic acid methyl ester (14 mg, 0.036 mmol, 1.2 eq, CAS:21085-72-3) and Ag_2_CO_3_ (10 mg, 0.036 mmol, 1.2 eq) were added to the vial and the mixture was then stirred at 80 °C. After 2 h the reaction mixture was cooled to room temperature and filtered using a 0.2 µm syringe filter and the filter was washed with 2 mL benzene. The resulting solution was concentrated with a stream of N_2_ gas. A ratio of 9:1 MeOH:1 M NaOH (1 mL) was added to the dried residue, and the resulting solution was stirred at room temperature. After 30 min the reaction mixture was concentrated with a stream of N_2_ gas, and the residue was lyophilized to yield a solid.

### 4.4. Genotyping

Participant genotypes were determined using whole genome sequencing as described in [19]. The combined vcf file was filtered to include only 520 samples with non-missing phenotypes. Further, variants on non-autosomes, variants with MAF <5%, genotyping call rate of <90%, and Hardy–Weinberg *p*-value <10^−6^ were filtered out leaving 6,375,079 variants for final analysis. All sample and variant filtering QC steps were performed using PLINK version 1.90b6.10 [40]. The indels were removed and variants were further pruned with LD threshold of r^2^ = 0.5 to include independent variants for the computations of principal components (PCs) and the relationship matrix. Ten PCs were used as covariates and estimated using PLINK, the relationship matrix was estimated using GCTA version 1.92.3 [41].

### 4.5. Locus Annotation

Genetic loci were annotated with nearest genes and overlapping eQTL, pQTL, mQTL, and methQTL information using PhenoScanner [20] and further refined using SNiPA [21] (Appendix A). Names of 54 genes related to the acetaminophen metabolism pathway were identified using GeneCards (N = 46, https://pathcards.genecards.org/card/acetaminophen_metabolism, accessed on 12 April 2022) and Drugbank (N = 21, https://go.drugbank.com/drugs/DB00316, accessed on 12 April 2022) and were then cross-referenced with the PhenoScanner annotation to identify candidate genes (Appendix A).

### 4.6. Statistical Analysis

All statistical analyses were conducted using R (version 4.1.0 and above) and Rstudio (version 1.4.1717 and above). A total of 45 ratios between metabolites were analyzed for association with all genetic variants and the p-gain was computed as described in [16]. The GWAS was conducted using the GCTA version 1.92.3 [42] with the mlma command using lognormal-scaled metabolite levels and pairwise ratios as dependent variables and age, sex, and the first ten PCs as covariates. The regional association plots were created using LocusZoom version 1.4 [43] using LD values from QBB participants.

## Figures and Tables

**Figure 1 metabolites-12-00496-f001:**
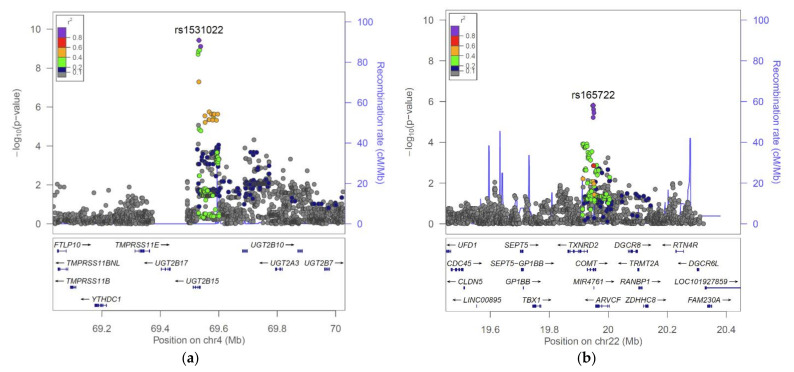
Regional association plots for the UGT2B15 (**a**) and the COMT (**b**) loci with the ratios 2-methoxyacetaminophen glucuronide*/2-methoxyacetaminophen sulfate* and 2-methoxyacetaminophen sulfate*/2-hydroxyacetaminophen sulfate*, respectively.

**Figure 2 metabolites-12-00496-f002:**
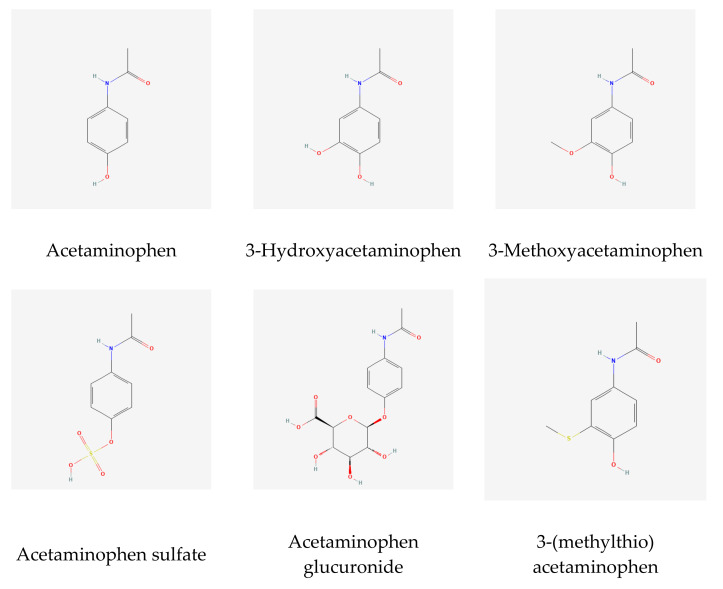
Chemical structures of key molecules discussed in this paper (images from PubChem https://pubchem.ncbi.nlm.nih.gov/, accessed on 15 March 2022).

**Figure 3 metabolites-12-00496-f003:**
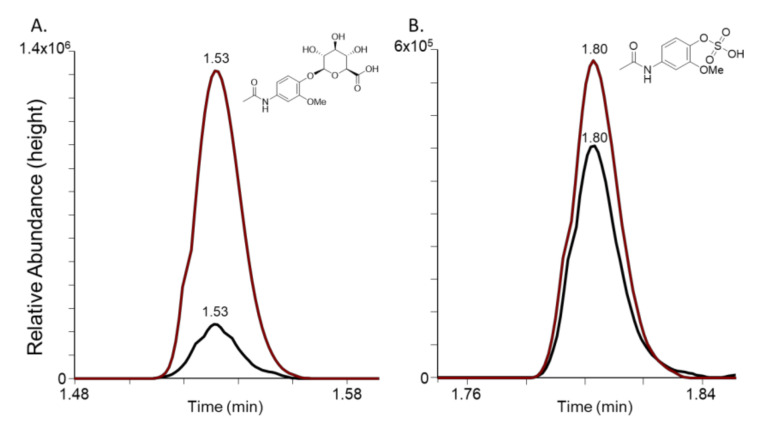
Co-elution of 3-methoxyacetaminophen metabolites (red) with peaks detected in human EDTA plasma (black). (**A**) Extracted Ion Chromatogram (XIC) of *m*/*z* 356.0987 +/− 5 ppm in negative ion mode for 3-methoxyacetaminophen glucuronide (red) and human EDTA plasma (black); and (**B**) XIC of *m*/*z* 260.02343 +/− 5 ppm in negative ion mode for 3-methoxyacetaminophen sulfate (red) and human EDTA plasma (black).

**Figure 4 metabolites-12-00496-f004:**
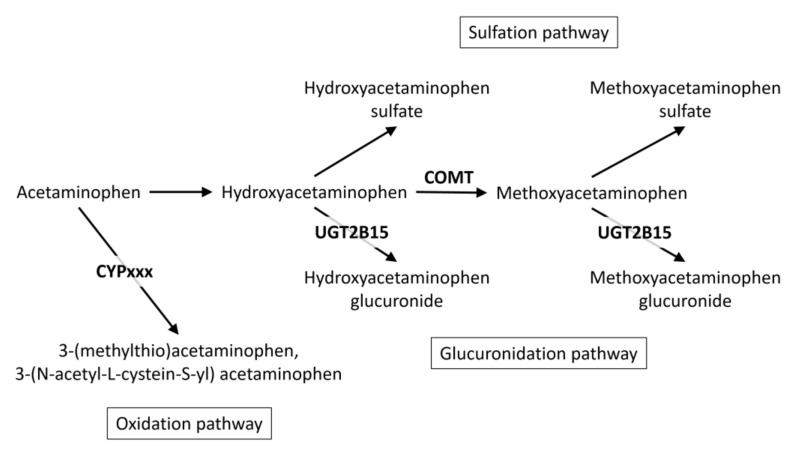
Schematic view of acetaminophen metabolism, showing metabolites and genes that were linked in this GWAS.

**Table 1 metabolites-12-00496-t001:** Nine acetaminophen metabolites detected in QBB.

BIOCHEMICAL	CAS	LC/MS Mode	Retention Index **	Mass	N (Out of 520)
4-acetamidophenol (paracetamol)	103-90-2	Neg	2173.7	150.05605	497
4-acetamidophenyl glucuronide	120595-80-4	Neg	1400	326.08814	510
2-methoxyacetaminophen glucuronide *	53446-12-1	Neg	1633	356.0987	416
2-hydroxyacetaminophen sulfate *	53446-14-3	Neg	1674	246.00778	497
2-methoxyacetaminophen sulfate *	53446-13-2	Neg	1949	260.02343	405
4-acetaminophen sulfate	10066-90-7	Neg	1792	230.01287	514
3-(N-acetyl-L-cystein-S-yl) acetaminophen	52372-86-8	Neg	2094	311.07072	363
3-(methylthio) acetaminophen sulfate *	78194-51-1	Neg	2265	276.00059	475
3-(cystein-S-yl) acetaminophen *	53446-10-9	Pos Early	2420	271.07471	363

* MSI level 2 molecule identifications, specifically identification based on spectral similarity to other known and fully characterized chemicals in a class, where commercial standards are not available for full confirmation [17]. Indeed, as we argue in this paper, the position of the hydroxy- and methoxy-groups of hydroxyacetaminophen and methoxyacetaminophen containing molecules is likely to be in the 3-position rather than in the 2-position as initially postulated based on literature precedence [18], i.e., 3-hydroxyacetaminophen sulfate (CAS 1702376-36-0), 3-methoxyacetaminophen sulfate (CAS 60603-12-5), and 3-methoxyacetaminophen glucuronide (52092-55-4), respectively. However, to avoid confusion please note that we did not rename these molecules in the Appendix A. ** The retention index is obtained by interpolation, relating the adjusted retention time of the sample component to the retention times of two standards eluted before and after the peak of the sample component, see [11] for details.

**Table 2 metabolites-12-00496-t002:** Summary statistics for the associations at the UGT2B15 and COMT loci.

Locus	Trait	SNP	Chr	Pos	MAF	Beta	*p*-Value	P-Gain
UGT2B15	2-methoxyacetaminophen glucuronide*/ 2-methoxyacetaminophen sulfate*	rs1531022	4	69,532,128	48.4%	0.150	3.7 × 10^−10^	4.6 × 10^7^
		rs72607822	4	70,331,309	5.8%	−0.253	1.2 × 10^−6^	1.7 × 10^4^
		rs7688257	4	68,126,425	8.5%	0.199	3.0 × 10^−6^	6.9 × 10^3^
		rs1968718	4	69,530,496	37.3%	−0.107	8.5 × 10^−6^	3.7 × 10^4^
COMT	2-methoxyacetaminophen sulfate*/ 2-hydroxyacetaminophen sulfate*	rs165722	22	19,949,013	45.0%	0.112	1.5 × 10^−6^	4.4 × 10^4^

## Data Availability

Access to Qatar Biobank data can be obtained through an established ISO-certified process by submitting a project request at https://www.qatarbiobank.org.qa/research/how-apply (accessed on 15 April 2022) which is subject to approval by the Qatar Biobank IRB committee.

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
