# Peer review of "Ratios of Acetaminophen Metabolites Identify New Loci of Pharmacogenetic Relevance in a Genome-Wide Association Study"

_metabolites, 2022, doi:10.3390/metabo12060496_

Round 1
Reviewer 1 Report
This manuscript presents a study of metabonomics using HPLC/HRMS and also GWAS methods to study a population and their metabolite excretion.
It is not clear to me if the blood sample come from 3000 people in the blood bank with unknown tratments and the about 500 were found having acetaminophen metabolites of if the patients were given acetaminophen? This should be explained better.
The authors found metabolites that are 3-methoxy-APAP derivatives. They though that this were 2-methoxy-derivatives according to a publication (ref 14 from 1974) However the 3-methoxy-APAP derivatives are known. I found 19 ref in Sci Finder and a complete description of these 3-Methoxy-APAP metabolites in a paper:
Formation and disposition of the minor metabolites of acetaminophen in the hamster. Gemborys MW, Mudge GH. Drug Metab Dispos. 1981 Jul-Aug;9(4):340-51. PMID: 6114834
But there are many other papers: Analysis of acetaminophen metabolites in urine by high-performance liquid chromatography with UV and amperometric detection. Wilson JM, Slattery JT, Forte AJ, Nelson SD. J Chromatogr. 1982 Feb 12;227(2):453-62. doi: 10.1016/s0378-4347(00)80398-9. PMID: 7061657
Cytochrome P-450 isozyme selectivity in the oxidation of acetaminophen. Harvison PJ, Guengerich FP, Rashed MS, Nelson SD. Chem Res Toxicol. 1988 Jan-Feb;1(1):47-52. doi: 10.1021/tx00001a009. PMID: 2979711
Hepatic protein arylation, glutathione depletion, and metabolite profiles of acetaminophen and a non-hepatotoxic regioisomer, 3'-hydroxyacetanilide, in the mouse. Rashed MS, Myers TG, Nelson SD. Drug Metab Dispos. 1990 Sep-Oct;18(5):765-70. PMID: 1981734
Advancing NMR sensitivity for LC-NMR-MS using a cryoflow probe: application to the analysis of acetaminophen metabolites in urine. Spraul M, Freund AS, Nast RE, Withers RS, Maas WE, Corcoran O. Anal Chem. 2003 Mar 15;75(6):1536-41. doi: 10.1021/ac026203i. PMID: 12659219
In addition the numerotation of the molecules should be explained since there is some ambiguity : What is the numerotation of acetaminophen (acetamino-phenol or paracetamol, para-acetylaminophenol (APAP , para, synonimous with 4-), 4-Acetamidophenol (here again phenol is leading). Or N-(4-Hydroxyphenyl)acetamide.
It looks that the 3-Methoxy-APAP compound has a CAS number: CAS : 52092-55-4
Your nomenclature suppose this formula for acetaminophen: APAP. You should state it somewhere.
I find the paper interesting.
The authors corrected a supposed error of the literature when in fact the literature has been corrected around 1980. After correcting these important literature errors and thus modifying there tables and text, (they could also state that they results were at odd with ref 14. Thus they did the synthetic and analytical work and then found the references to the newly synthetised compounds.)
Anyhow the study demonstrate that some unknown metabolites can be revealed and that they can be correlated to genome variations on metabolic pathways.
Thus this is an interesting finding. The GWAS part seems well done. The HPLC/MS too.
I think the publication should be modified according to the preceeding remarks. The reference 14 was misleading. But general drug metabolism knowledge should have corrected this. (I teach the pathway described by Nelson since the 1980. Thus my students should know.)
I think that the paper need a major correction. I will join a Scifinder search on the 3-methoxy metabolite.)
Reviewer 2 Report
- Some jargon information shown in Table 1 might confuse readers. For example, what is the unit and the transformation of the retention index. If possible, the author should consider more information in the footnote.
- A metabolic pathway with the associated enzymes would be easier to explain the ratio of the compounds relationship instead of the structures illustrated in figure 2. And/or one structural change-related figure might be a good explanation as well.
